# Antiviral Activities of HIV-1-Specific Human Broadly Neutralizing Antibodies Are Isotype-Dependent

**DOI:** 10.3390/vaccines10060903

**Published:** 2022-06-06

**Authors:** Blandine Noailly, Melyssa Yaugel-Novoa, Justine Werquin, Fabienne Jospin, Daniel Drocourt, Thomas Bourlet, Nicolas Rochereau, Stéphane Paul

**Affiliations:** 1CIRI—Centre International de Recherche en Infectiologie, Team GIMAP, Université Claude Bernard Lyon 1, Inserm, U1111, CNRS, UMR530, CIC 1408 Vaccinology, 42023 Saint-Etienne, France; blandine.chanut@univ-st-etienne.fr (B.N.); melyssa.yaugel.novoa@univ-st-etienne.fr (M.Y.-N.); justwerquin@gmail.com (J.W.); fabienne.jospin@univ-st-etienne.fr (F.J.); thomas.bourlet@chu-st-etienne.fr (T.B.); rochereau.nicolas@gmail.com (N.R.); 2Cayla InvivoGen, 31400 Toulouse, France; e.perouzel@invivogen.com

**Keywords:** HIV, broadly neutralizing antibodies, gp41, gp120, ADCC, neutralization

## Abstract

Broadly neutralizing antibodies (bNAbs) offer promising opportunities for preventing HIV-1 infection. The protection mechanisms of bNAbs involve the Fc domain, as well as their Fab counterpart. Here, different bNAb isotypes including IgG1, IgA1, IgA2, and IgA122 (IgA2 with the hinge of IgA1) were generated and then produced in CHO cells. Their ability to neutralize pseudovirus and primary HIV-1 isolates were measured, as well as their potential ADCC-like activity using a newly developed assay. In our work, gp41-specific IgA seems to be more efficient than IgG1 in inducing ADCC-like activity, but not in its virus neutralization effect. We show that either gp120-specific IgA or IgG1 isotypes are both efficient in neutralizing different viral strains. In contrast, gp120-specific IgG1 was a better ADCC-like inducer than IgA isotypes. These results provide new insights into the neutralization and ADCC-like activity of different bNAbs that might be taken into consideration when searching for new treatments or antibody-based vaccines.

## 1. Introduction

Immunoprophylaxis with potent bNAbs efficiently protects non-human primates from mucosal transmission even after repeated challenges [1,2]. However, the precise mechanisms of bNAb-mediated viral inhibition in mucosal tissues are currently under-investigated. Constant heavy-chain domains, such as the CH1 domain, modify antibody affinity and specificity, which demonstrates that not only variable regions contribute to antigen binding [3,4,5,6,7,8,9]. Increasing evidence shows that non-neutralizing Abs play a significant role in decreasing the viral load, leading to partial and sometimes even total protection. The mechanisms suspected to participate in protection involve the Fc domain of Abs, as well as their Fab counterpart. Consequently, the induced Ab isotype will be determinant for their functions, as well as the quantity and quality of the Fc receptors (FcRs) expressed on immune cells. Fc-mediated inhibitory functions, such as Ab-dependent cellular cytotoxicity (ADCC), antibody-dependent cellular phagocytosis (ADCP), aggregation, and even immune activation, have been proposed [4,5,10,11,12,13]. It has been demonstrated that longer hinges are important in the phagocytic activity of IgG1 and IgG3 isotypes, but not for ADCC or neutralization [14]. Recently, Duchemin et al. compared IgA2 and IgG1 isotypes from the 2F5 antibody and showed that the IgA2 isotype induced better ADCC and ADCP effects than the IgG1 isotype [15,16]. Moreover, Jia et al. in 2020 isolated, from chronically infected donors, two bNAbs B cell lineages that class-switched to IgG and IgA. IgA variants reconstituted from these bNAbs had broadly neutralizing activity as well as plasma IgA fraction [17]. These recent findings highlight the importance of IgA- and IgG-related Fc-effector functions. The RV144 HIV vaccine clinical trial identified plasma IgA responses to HIV Env as correlating to increased HIV acquisition and a decrease in the in vitro ADCC activity of vaccine-induced Env-specific IgG [18]. 

Here, we conducted a deeper analysis of the isotype impact on the function of bNAbs targeting the gp120 and gp41 regions of HIV-1 Env. The increased interest in understanding the role of ADCC in preventing and controlling HIV-1 infection leads us to assess this function of a panel of antibodies [11]. We studied a panel of Env-specific bNAbs (2F5, B12, PG16, PGT126, PGT128, PGT145, NIH45-46, and 10E8 (Table 1)) for which the ability to mediate non-neutralizing functions is not completely described. IgG1, IgA1, IgA2, and IgA122 (IgA2 with the hinge of IgA1) for the 2F5 bNAb family; IgG1, IgA2, and IgA122 for B12; and IgG1 and IgA122 for the rest of the bNAbs were used to compare their ability to neutralize pseudoviruses and primary HIV-1 isolates, as well as their potential ADCC activity in a new potent gp120- and gp41-targeted assay.

## 2. Materials and Methods

### 2.1. Cells 

TZM-bl and CHOgp140 cell lines were obtained from the NIH AIDS Reagent Program. HEK 293-gp41MSD (Membrane Spanning Domain) has been previously described [19]. CHO-gp140 and HEK 293-gp41 express gp140 or gp41 in its trimeric form. HEK CD89^+^ and HEK CD16^+^ cell lines were obtained from InvivoGen.

### 2.2. Construction and Production of Recombinant IgA1, IgA2, IgA122, and IgG 

CHO cell lines were transfected with two plasmids: pFUSE-CHIg and pFUSE2-CLIg. These plasmids express the constant regions of the heavy (CH) and light (CL) chains, respectively. Transfection of the CHO cell lines with the recombinant pFUSE-CHIg and pFUSE2-CLIg pair allows the generation of an Ig antibody. Antibodies were purified from the supernatant using the appropriate protein affinity chromatography. IgG1 was purified with protein G, and IgA with peptide M. IgA122 was obtained by replacement of the IgA2 hinge region with the IgA1 hinge region. The purity of recombinant Abs was analyzed on SDS-PAGE in reducing conditions, and the gel was stained with Coomassie Blue.

### 2.3. Analysis of the Specificity of the Different bNAb Isotypes

Target cells were incubated with broadly neutralizing antibodies (bNAbs) IgA1, IgA2, IgA122, or IgG1 at 5 µg/mL, followed by an anti-human IgGAM (fluorescein isothiocyanate (FITC); Abliance). Antibody binding to target cells was quantified by flow cytometry using a CantoII flow cytometer (BD), and the geometric mean fluorescence intensity (gMFI) was calculated using the software FlowJo, version 10.6.2 (BD). 

### 2.4. HIV-1-Specific Neutralization Assays

Viral primary isolates of clades A (92UG029), B (SF162, 92US660), and D (92UG001) were obtained from the NIH AIDS Reagent Program. Pseudoviruses were produced by co-transfection of the backbone plasmid PSG3Δenv (NIH) and the envelope plasmid of clade B (11035, 11036, and 11307) or clade C (11312) (NIH) in the HEK293 cell line. 

### 2.5. TZMbl/Pseudovirus Assay

Pseudoviruses were incubated with the bNAbs at different concentrations for 2 h, and these mixtures were added to the cells (10,000 cells/well) plated in 96-well plates for 24 h. After 48 h of infection, luciferase activity was used to quantify the infected TZM-bl cells by adding the luciferase substrate, Bright Glo (Promega), and the luminescence was measured (Berthold Technologies ref Tristar LB941). Neutralization activity directly correlates with the reduction in luciferase reporter gene expression after a single round of virus infection in TZMbl cells by the pseudoviruses of panels B and C. 

### 2.6. PBMC/Primary Isolate Assay

Neutralization assay for the primary isolates of clades A (92UG029), B (SF162, 92US660), and D (92UG001) was performed using a pool of PBMCs isolated from healthy donors and stimulated with Phytohemagglutinin (PHA) (2 µg/mL) and recombinant human IL-2 (R&D Systems 202IL) (200 U/mL). A dose of 100 TCID50 of primary isolate virus were incubated for 2 h with the bNAbs at different concentrations, and with these mixtures, PBMCs were infected for 3 h at 37 °C. The amount of p24 in the supernatants was quantified using an ELISA (InfYnity biomarkers). The percentage of neutralization and EC50 were calculated as previously described [20].

### 2.7. HIV-1 Env-Dependent Antibody-Dependent Cellular Cytotoxicity (ADCC)-like Assay

ADCC-mimicry assay was performed using cells expressing HIV-1 Env (HEKgp41 or CHOSEC/CHOgp140) [21] as target cells. Target cells were first incubated with different bNAb concentrations. Effector cells expressing the Fc receptor (HEK-CD16^+^ or HEK-CD89^+^) were first established by transfection (Invivogen, data not shown). HEK-CD16^+^ or HEK-CD89^+^ cell lines used in the ADCC-mimicry assay have the same expression level of CD16 or CD89 measured by flow cytometry (Invivogen). HEK-CD16^+^ or HEK-CD89^+^ cell lines co-incubated with bNAbs were co-cultured with the target cells at a 1:1 effector/target ratio for 48 h at 37 °C. Alkaline phosphatase was measured in the culture supernatant using Quantiblue (InvivoGen) at 620 nm. Percentage of ADCC was calculated using the formula: 100× (OD 620 nm with bNAb—OD 620 nm without bNAb)/OD620 nm with bNAb.

### 2.8. Statistical Analysis

All statistical analyses were performed with InStat software (version 8.01; GraphPad Software, La Jolla, CA, USA). The normality was tested for each data set with both Shapiro–Wilk and D’Agostino–Pearson tests. When data could be modelled by a normal distribution, the following tests were performed depending on the comparison: Student’s *t*-test (comparison of two means), one-way ANOVA (multiple mean comparisons) with Bonferroni correction (comparison of all pairs of data sets) or Dunnett’s correction (comparison to a control group). Alternatively, non-parametric tests were performed with the Mann–Whitney test (comparison of two means). In the case of comparisons of two independent variables, a two-way ANOVA test was done. Results are represented as Mean ± SEM. *p* values of less than 0.05 (*), less than 0.01 (**), and less than 0.001 (***) were considered significant. Statistically significant differences between groups are emphasized by bars connecting the relevant columns. 

## 3. Results

### 3.1. Development of a New Highly Potent Cellular Assay to Measure the ADCC-like Activity

We used a new method able to monitor the potential ADCC-like activity of different mAbs using HEK 293-gp41MSD and CHOgp140 cells as target cells [19,22]. Different HIV-bNAbs from different isotypes (IgA1, IgA2, and IgG1) were generated to compare their functional properties. An antibody IgA122, using an IgA2 Fc with the hinge region of IgA1, was also interestingly included to investigate whether the hinge length can affect the effector function of IgA isotypes. The four isotypes were produced for the bNAb 2F5 and IgG1, IgA2, and IgA122 for B12, while for the rest, we constructed IgG1 and IgA122. The antibodies were generated by recombinant DNA technology, produced in CHO cells, and purified by affinity chromatography according to their isotype (protein G for IgG, peptide M for IgA), obtaining from 2–5 mg of each antibody with more than 98% purity. We could identify the presence of the heavy (H) and light (L) chains at the expected sizes, which proves the integrity of each antibody (Figure 1). 

The binding capacity of the bNAbs to their respective antigens was first tested by flow cytometry on the two target cell lines. IgG1 recognized its epitopes on gp41 and the CD4 binding site in the gp120 protein (MFI = 407–6784) better than the IgA isotypes (MFI = 152–1548). The V2 and V3 epitopes were similarly unrecognizable by all isotypes (Table 1). All tested antibodies exhibited an ADCC-like dose effect ranging from 0.75 µg/mL to 100 µg/mL (Figure 2). For most of the antibodies, the ADCC-like maximal activity effect was clearly achieved at the maximum tested concentration.

### 3.2. ADCC-like Activities of bNAbs Depend on Their Isotypes and Their Specificity

#### 3.2.1. Higher ADCC-like Activity of gp41-Specific IgA Isotypes

For 2F5, the ADCC-like activities of IgA122 (EC50 = 25.40 µg/mL), IgA1 (EC50 = 30.13 µg/mL), and IgA2 (EC50 = 19.51 µg/mL) were statistically different (*p* < 0.0001) from IgG1 ADCC (EC50 = 100 µg/mL) (Figure 2). However, no significant differences were observed between the IgA isotypes. For 10E8, the IgA122 isotype also had more potent ADCC-like activity (EC50 = 14.94 µg/mL) than the IgG1 isotype (EC50 = 58.49 µg/mL) (*p* < 0.000001) (Figure 2). The 2F5 and 10E8 antibodies have their epitopes in the MPER region of gp41. However, 2F5 recognizes a beta sheet epitope and 10E8 an α-helix epitope [23]. Nevertheless, our results suggest that, in the case of 2F5 and 10E8, the IgA isotypes are better ADCC inducers than IgG1 against gp41, independently of the nature of the epitope.

#### 3.2.2. Heterogenous gp120-Specific ADCC-like Activity of IgG1 and IgA Isotypes 

Measurements of the ADCC-like activity of gp120-specific bNAbs were tested on CHOgp140 target cells (Figure 2). When comparing the NIH4546 and B12 families, which have their epitopes in the CD4 binding site, we detected that, in the case of NIH4546, IgG1 induced 50% ADCC with a higher antibody concentration (EC50 = 2.61 µg/mL) than IgA122 (EC50 = 1.569 µg/mL) (*p* = 0.00095). In the case of B12, IgA122 needed more concentration to achieve 50% ADCC (EC50 = 31.80 µg/mL) than IgA2 (EC50 = 14.22 µg/mL) and IgG1 (EC50 = 15.87 µg/mL), and the differences were statistically significant in both cases (*p* < 0.0001). However, the EC50 concentrations of the IgG1 and IgA2 isotypes on CHOSEC were not statistically different (*p* = 0.48). For PGT128, no significant differences between 50% ADCC-like activities mediated by the IgA122 and IgG1 isotypes were observed (IgA122 EC50 = 100 µg/mL; IgG1 EC50 = 86.38 µg/mL; *p* = 0.558). Interestingly, for PGT126-specific isotypes, we observed a significant decrease in IgA122 EC50 concentrations compared to IgG1 (IgA122 EC50 = 7.81 µg/mL; IgG1 EC50 = 46.62 µg/mL; *p* < 0.000001). In contrast, for PGT145, IgG1 had a significantly higher ADCC activity than IgA122 with less antibody concentration (IgA122 EC50 = 100 µg/mL; IgG1 EC50 = 20.52 µg/mL; *p* = 0.0004). For PG16 isotypes, the IgG1 EC50 antibody concentration was significantly smaller than IgA122 (IgA122 EC50 = 100 µg/mL; IgG1 EC50 = 2.32 µg/mL; *p* = 0.00003). The IgA and IgG1 isotypes induced a heterogeneous ADCC effect when facing cells expressing the gp120 glycoprotein. Of note, for V2-specific antibodies, IgG1 induces more ADCC, probably due to the better accessibility of IgG1 than IgA in the V2 glycan shield structure [24,25]. 

### 3.3. Development of a New Highly Potent Cellular Assay to Measure the ADCC-like Activity

#### 3.3.1. Gp41-Specific IgG1 Has More Potent Neutralizing Activity

When comparing the role of the bNAbs’ Fc portion in their virus neutralization ability, any statistically significant difference between bNAbs was, measured. However, for gp41-specific antibody families as previously described, we confirmed the superiority of 10E8 (Figure 3), implying that the slightly greater access of the 10E8 bNAb to the MPER epitope on the cell surface than that of 2F5 may play a critical role in antibody neutralization, as was already shown by Huang et al. in 2012 [26]. 

When we compared isotypes, IgG1 from the 10E8 and 2F5 bNAbs better neutralized six out of eight viral strains with an EC50 < 7 µg/mL, while the IgA isotypes neutralized one or two viral strains (92UG001 strain for IgA2 2F5 antibody and 92UG029 and SF162 strains for IgA1 2F5; EC50 < 0.02 µg/mL). Only IgA122 from 10E8 neutralized the four pseudoviruses tested (EC50 < 8.06 µg/mL), suggesting that, for this specific family, the hinge length increases the neutralization ability in that case.

#### 3.3.2. Gp120-Specific IgG1 and IgAs Have Similar Neutralization Effect

As for the gp41-specific antibodies, we did not find any statistically significant differences between the neutralization effects of any of the gp120-specific tested bNAbs. Nevertheless, we saw that only B12 IgA122 and NIH4546 IGA122 and IgG1 neutralized all viral strains with an EC50 < 13 µg/mL. For the remaining tested bNAbs, the IgG1 and IgA isotypes similarly neutralized all viral strains, even though the pseudoviruses were better neutralized than the primary isolates (Figure 3). The IgA122 isotype tends to better neutralize all viruses than the respective IgG1 isotype. That is the case for PG16 family, where the IgA122 isotype neutralizes six out of eight viruses with an EC50 < 4 µg/mL, and the IgG1 isotype also neutralizes six out of eight viruses, but with a higher antibody concentration (EC50 < 14 µg/mL). The B12 family is also an example of that because, even if the IgA1 and IgA2 isotypes were exclusively better neutralizers of pseudoviruses and IgG1 had a broader spectrum of neutralization, IgA122 required less antibody concentration to have the same effect (EC50 < 16 µg/mL) against all tested viruses.

### 3.4. Correlation between ADCC-like Activity and Virus Neutralization

Each of the HIV-1 monoclonal antibodies was tested for neutralization by two methods with different viruses (clades A, B, and D) or pseudoviruses (clades B and C). ADCC-like activity was also measured and correlated with antibody binding to target cells (Figure 4a). For gp41, the ADCC-like EC50 (Spearman’s r = 0.6) tended to positively correlate with antibody binding to target cells; however, the correlation was not significant (*p* > 0.4) (Figure 4a). Neither did the ADCC-like EC50 significantly correlate with neutral activity for any of the eight viral strains evaluated (Spearman’s r = 0.20; 0.03; 0.34; −0.58; 0.52; −0.70; −0.67; −0.70; *p* > 0.1) (Figure 4b). However, there is a clear tendency to a better ADCC-like effect for the IgA isotypes when more potent neutralization is achieved, even though this effect depends on the viral strain used for the neutralization test. On the other hand, the IgG isotype tends to be better for neutralization and less efficient for ADCC, compared to IgA isotypes. For gp120-specific IgA and IgG, the ADCC-like activity inversely correlated with binding to target cells (Spearman’s r = −0.82; −0.09, respectively) (Figure 4a). Both isotypes seem to be equally efficient for neutralization; nevertheless, IgG tends to generate a better ADCC response than the IgA isotypes. A positive correlation between the ADCC and neutralization of the 92UG001 viral strain was observed when comparing the IgA isotypes (Spearman’s r = 0.88, *p* = 0.0179) (Figure 4b).

## 4. Discussion

Here, we compare the impact of different isotypes of bNAbs on their antiviral functions. The evaluation of the potential ADCC of different isotypes from different bNAbs will provide information concerning the contribution of the Fc region of gp41-specific and gp120-specific antibodies in the protection against HIV-1. For that reason, the IgG and IgA isotypes from eight bNAb families were constructed and produced in CHO. The quality of these antibodies was evaluated by their specificity and binding capacity to CHOgp140 or HEK-gp41 target cells; however, IgA was less efficient in binding gp41- and gp140-expressing cells compared to the IgG isotypes. We then tested their potential ADCC-like activity and their ability to neutralize primary isolates and pseudoviruses. ADCC-like activity was measured using a new sensitive assay designed to measure phosphatase alkaline, from which the expression was enhanced by the binding between FcR-expressing cells and the complex Ab-target cell. Ab antiviral efficiency is highly dependent not only on the Fab–antigen interactions that block viral entry but also on interaction of the Ab Fc domain with its cognate FcR expressed on the innate effector cells [27]. Contrary to other ADCC tests using PBMCs or NKs as effector cells [15,16,28], the assay is simpler and more interesting to gain an idea of the potency of the Fc-dependent antiviral effects that bNAbs can trigger beyond ADCC. However, further studies are needed to compare our assay with a more physiologic assay using PBMCs or NK cells. Our test uses antibody concentrations higher than the ones used by Duchemin et al. in 2020 [16]. Nevertheless, with this new method for evaluating ADCC effect based on cells expressing gp41 or gp120, all bNAbs tested were able to induce an ADCC response without the need to work with HIV-1-infected cells.

Our study reveals that, in the context of gp41, despite the inverse correlation between the ADCC-like EC50 and the effective binding of IgG1 to target cells, IgA is more efficient than IgG1 in inducing ADCC, but not for its neutralization effects in the context of the virus. In accordance with this, the IgG1 isotypes from 2F5 and 10E8 are described to have less or no ADCC effect against HIV-1-infected cells [29], probably due to a low affinity of these Abs for Env protein in the context of HIV-1-infected cells, which can be explained by their specificity to a gp41 epitope that is transiently exposed during fusion and phospholipids [30,31,32,33]. Moreover, Tudor et al. and Duchemin et al. showed that the IgA2 isotype from the 2F5 bNAb induced more ADCC and ADCP than its corresponding IgG1 [4,16]. For gp120 specificity, the IgA or IgG1 isotypes are efficient in neutralizing the different viral strains. In contrast, IgG1 was a better ADCC inducer than the corresponding IgA isotypes. The differences found between the Fc effector functions depending on the target protein could be explained by the different structural conformations between the Env glycoprotein on the infected cells and on the virus, making the epitopes variably exposed to the Ab. In accordance with this, Benjamin von Bredow et al. in 2019 showed for the PGT145 Ab that antibody binding affinity can differentiate sensitivity to ADCC from neutralization [34].

Duchemin et al. demonstrated that, for the gp41-specific 2F5 bNAb, the IgA isotype cooperates with the corresponding IgG isotype and also with IgG from 10E8 to increase HIV-1-infected cell lysis by ADCC [15]. However, Tomaras et al. addresses the interference of gp120-specific IgA in the correspondent IgG-mediated ADCC, most likely due to the higher affinity of IgA than IgG for gp120 [35]. In this last study, they used effector cells that did not express the FcαRI and were unable to lyse target cells mediated by IgA isotypes. Duchemin et al. proposed that one possible explanation of the differences between their study and Tomaras et al. is that the capacity of the gp120 epitopes targeted by the RV144 vaccine-induced Abs in mediating ADCC differs from that of the gp41 epitopes. The results of our study support their conclusions, suggesting that the epitopes targeted by the Ab is crucial for isotype cooperation in ADCC. 

Khamassi et al. in 2020 showed that the CH1α domain of FabA has strong antibody specificity and functional activities, notably, an efficient neutralization of HIV-1-infected CD4+T lymphocytes, which is not true for FabG [36]. The same group saw a gain in affinity and antiviral activity of bNAbs when they performed isotype switching from IgG to IgA [4,36]. Their studies add another layer of complexity to the structure–function relationship of bNAbs, and our results are in accordance with this, as we detected differences between IgA and IgG isotypes’ antiviral functions for some families. On the other hand, the production of Ab IgA2 containing the hinge of IgA1 showed that there is no gain in the Fc effector functions tested in this study for the bNAb 2F5. These findings contrast with Richardson et al. in 2019 [10] showing that the hinge region and Fc portion of a bNAb influenced the neutralization and Fc-relative functions. However, for the B12 family, the isotype IgA122 better neutralized pseudoviruses and primary isolated viruses than the other IgA B12 isotypes. At the same time, IgA2 was a better neutralizer than IgA1 for the same family. It seems that, for the B12 antibody, a strategy of recombinant antibodies with the Fc part corresponding to IgA2 and the IgA1 hinge could be a good approach for immunotherapies against HIV. In addition, Scheepers et al. showed that IgA1 switching from IgG1 improved the neutralization activity of CAP88-CH06 lineage antibodies, and the suggested mechanism was associated to the structure and glycosylation of the IgA1 hinge region [37]. In accordance with the greater exposure of the Env proteins of lab-adapted viruses to antibodies [38,39,40,41], pseudoviruses were generally more sensitive to neutralization in terms of the magnitude of responses; notwithstanding, some antibodies were more efficient in neutralizing the primary viruses than the pseudoviruses. 

This study has limitations. The test used here to measure ADCC-like activity does not reflect the physiological ADCC that occurs in the body, as there are no effector cells, such as PBMCs or NKs. However, it offers an idea of the potency of the Fc-dependent antiviral effects that bNAbs can trigger beyond ADCC. Nevertheless, further studies are needed to compare our assay with a more physiologic assay using PBMCs or NK cells.

Overall, our results reveal a variety of responses according to the nature of the neutralized virus and the Env-specific ADCC-related effect for the bNAbs tested. Concerning the ADCC-like effect of the gp41-specific tested antibodies, the IgA isotypes seem to be better than IgG1; the contrary seems to be true for the gp120-specific tested antibodies. We consider that the isotype selection should be based on the target epitope and the desired Fc-related effect. These results provide new insights into the neutralization and ADCC effects of different bNAbs that might be taken into consideration when searching for new treatments or antibody-based vaccines. 

## Figures and Tables

**Figure 1 vaccines-10-00903-f001:**
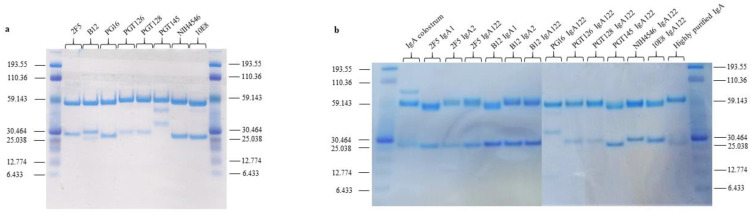
Characterization of the different bNAb isotypes: (**a**) Separation of bNAbs IgG and (**b**) IgA isotypes on SDS-PAGE (Bis-Tris 4–12%). An amount of 10 µg of the corresponding antibody was loaded on the gel. The first and last lanes of each gel correspond to the molecular weight marker. Two IgA controls were used: colostrum IgA and a highly purified IgA. Migration was performed at 150 V for 1 h.

**Figure 2 vaccines-10-00903-f002:**
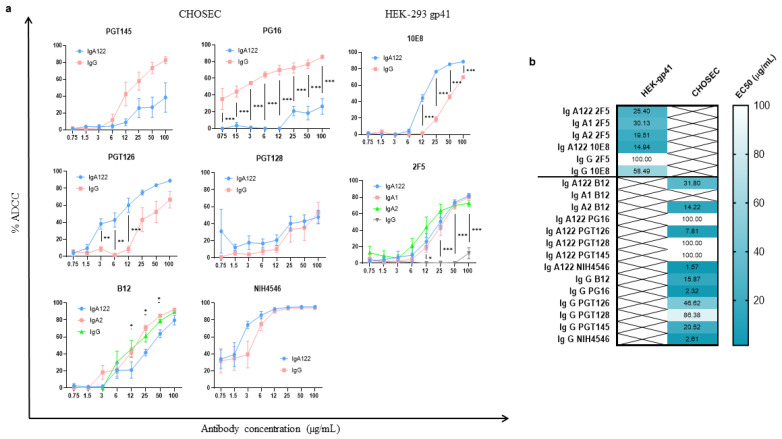
ADCC-like activities of bNAbs depend on their isotypes and their specificity: (**a**) %ADCC for each evaluated concentration. Each point represents the mean and SEM of triplicates from two different experiments. Student’s t-test was used for antibodies PGT145, PG16, 10E8, PGT126, PGT128, and NIH4546. A 2-way ANOVA with Tukey’s multiple comparisons was used for the antibodies 2F5 and B12. * *p* < 0.05, ** *p* < 0.01, *** *p* < 0.001 (**b**) Heatmap of EC50 (µg/mL) values obtained for ADCC. Numbers indicate the geometric mean values of results from triplicates of at least two independent experiments (n = 6).

**Figure 3 vaccines-10-00903-f003:**
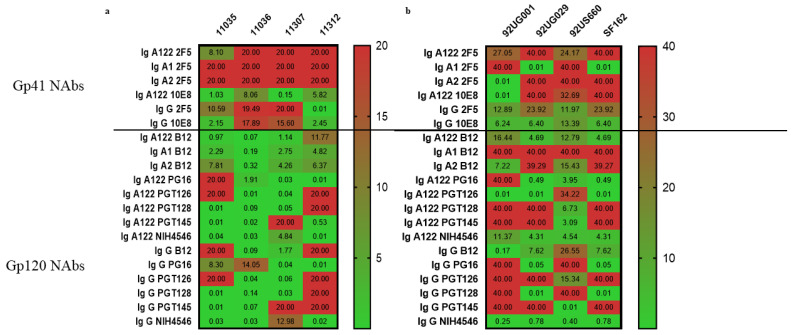
Neutralizing activity of bNAbs depend also on their isotypes and their specificity. Heatmap of EC50 (µg/mL) values obtained for neutralization for all tested antibodies using: (**a**) TZM-bl cells and pseudovirus or (**b**) PBMCs isolated from healthy donors and primary isolates. Numbers indicate the geometric mean values of results from duplicates of at least two independent experiments (n = 4). A 2-way ANOVA with Tukey’s multiple comparisons was used to compare EC50s from all the antibodies per virus.

**Figure 4 vaccines-10-00903-f004:**
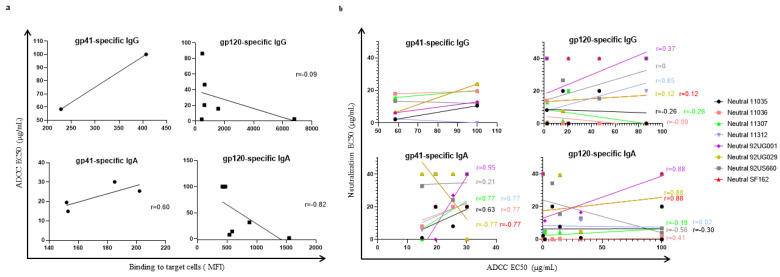
Correlation between: (**a**) MFI and ADCC-like EC50, (**b**) ADCC-like EC50 and Neutralization EC50. Each point represents the mean value of triplicates from two different experiments (n = 6). Non-parametric Spearman correlation was calculated for each case.

**Table 1 vaccines-10-00903-t001:** Efficacy of binding of the different isotypes on Env-expressing cells by flow cytometry. Percentage of positivity of cells with 5 µg/mL of antibodies (n = 3).

Target Protein	Epitope *	Antibody	Isotype	MFI	% of Positivity
gp41 (HEK-gp41)	MPER	2F5	IgG1	407	93.5%
			IgA1	185	55.0%
			IgA2	152	45.5%
			IgA122	202	64.5%
		10E8	IgG1	228	41.6%
			IgA122	153	8.76%
gp140 (CHOSEC)	CD4 binding site	B12	IgG1	1566	99.6%
			IgA1	886	98.4%
			IgA2	586	73.5%
			IgA122	878	92.3%
		NIH4546	IgG1	6784	100%
			IgA122	1548	99.4%
	V2 glycan	PG16	IgG1	439	0.049%
			IgA122	422	0.23%
		PGT145	IgG1	623	41.4%
			IgA122	471	4.34%
	V3 glycan	PGT126	IgG1	638	47.6%
			IgA122	540	27.4%
		PGT128	IgG1	476	13.2%
			IgA122	442	1.62%

* Source: NIH AIDS Reagent Program; https://www.hiv.lanl.gov/ (accessed on 9 December 2021).

## Data Availability

Not applicable.

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
