# Peer review of "Antiviral Activities of HIV-1-Specific Human Broadly Neutralizing Antibodies Are Isotype-Dependent"

_vaccines, 2022, doi:10.3390/vaccines10060903_

Round 1
Reviewer 1 Report
The manuscript by Noailly et al describes isotype dependent antiviral activities of HIV1-specific human broadly neutralizing antibodies (bNAbs). Different isotypes as IgG1, IgA1, IgA2 and IgA122 (IgA2 with the hinge of 13 IgA1) were tested for their ability to neutralize and potential ADCC activity using a new developed assay. The experiments are straightforward. Overall, the results are extension of previous studies perhaps with limited new information. The new assay system could help for initial screening system but may not be critical for making crucial conclusion.
This manuscript could be enhanced significantly if there was comparable study of the new assay with biological relevance cells.
The author described they used higher antibody concentration compared to previous studies. Should their study include different antibody concentrations?
Should there be a structural different expressed env vs the env on virus particles? What control experiment was included for this potential difference?
Author Response
The manuscript by Noailly et al describes isotype dependent antiviral activities of HIV1-specific human broadly neutralizing antibodies (bNAbs). Different isotypes as IgG1, IgA1, IgA2 and IgA122 (IgA2 with the hinge of 13 IgA1) were tested for their ability to neutralize and potential ADCC activity using a new developed assay. The experiments are straightforward. Overall, the results are extension of previous studies perhaps with limited new information. The new assay system could help for initial screening system but may not be critical for making crucial conclusion.
Thank you for your appreciation of our work.
This manuscript could be enhanced significantly if there was comparable study of the new assay with biological relevance cells.
This is a very good point highlighted by the reviewer. We did not have the opportunity to correlate a conventional ADCC test with a direct measure of cytotoxicity. We no longer have access to anti-HIV antibodies because InvivoGen has stopped making them for us and our old stock is now contaminated. In order to limit confusion, we propose to change the name of our assay to ADCC-mimicry assay or ADCC-like assay. ADCC has been replaced by ADCC-like assay throughout the manuscript.
The author described they used higher antibody concentration compared to previous studies. Should their study include different antibody concentrations?
This is an important point. All tested antibodies exhibited a dose effect ADCC-like ranging from 0.75 µg/mL to 100 µg/mL (Figure 2). For most of the antibodies, the ADCC-like maximal activity effect was clearly achieved at the maximum tested concentration.
Should there be a structural different expressed env vs the env on virus particles? What control experiment was included for this potential difference?
The cell lines used for the tests were all validated using conformational antibodies recognizing the envelope (gp120 or gp41) in its native environment (described extensively in references 19 and 22).
Reviewer 2 Report
This is an interesting article that makes the point that broadly neutralizing antibodies (BNAbs) against HIV-1 may differ in their ability to neutralize virus and mediate ADCC based on Fc and hinge region characteristics in addition to epitope recognition and binding by the Fab region. Furthermore, the impact of specific Fc and hinge region on antibody functionality may vary depending on the Fab specificity. This is not a surprising finding as many other studies have shown instances of such differences, however this study provides greater detail in a broader survey of BNAb specificities presented with a variety of FC and hinge regions. More importantly, this paper is the first presentation of a novel ADCC assay that may be easier to perform than standard such assays. Unfortunately, there is one major flaw and several minor flaws in the paper that make it unsuitable for publication in its current form.
The major flaw has to do with the description of the new method and is actually noted by the authors themselves in the discussion where they state “This study has limitations. The test used here to measure ADCC do (sic) not reflect the physiological ADCC that occurred in the body, as there are no effector cells like PBMCs or NKs. … further studies are needed to compare our assay with a more physiologic assay using PBMCs or NK cells.” They state in Materials and Methods (2.6 the description of the new assay) that “Ability of HEK-CD16+ or HEK-CD89+ to mimic ADCC were compared to classical NK-based ADCC assay” but then no data or reference to published data for the comparison is given. They further state in Results (3.1) that “We previously defined a new method able to monitor the potential ADCC activity of different mAbs using HEK 293-gp41MSD and CHOgp140 cells as target cells” and give two references for that statement; but neither reference is to a paper that even mentions the new method (one is a reference to the HEK 293-gp41MSD cell line and the other is a totally unrelated article that just happens to be co-authored by three of the authors on this paper). As the first report of a new assay it is unacceptable to not include any data comparing the new assay with standard assays, especially if the authors themselves realize this limits the paper and actually have the data as they appear to indicate. The fact that it is unclear what relationship this new assay bears to ADCC measured by standard assays means that the results reported with the new assay are of unclear relevance to designing monoclonal antibodies for use as therapeutic agents or for prophylaxis; this negates the value of the work done. On a minor note, the relationship between ADCC directed at gp41 epitopes in intact virus (where gp120 restricts access by an antibody bound to the surface of a cell) and ADCC directed at gp41 presented on the HEK-gp41 cell line (where the missing gp120 does not restrict access to these difficult to reach epitopes) is shaky at best and may not reflect the differing usefulness of the MAbs with different FC or hinge regions.
Another problem with the paper is the data itself, much of which represents trends that do not actually meet statistical significance.
Lastly, some of the language in the is awkward and sounds like a non-native speaker of English used google to translate what was written in another language; this paper would benefit from being proofread by someone who is both a native speaker of English and knowledgeable about the science.
In summary, this is an interesting paper but lack of data comparing the novel assay presented with more traditional ADCC assays severely compromises the analysis and possible relevance of the findings. The authors repeatedly suggest that they have made such a comparison – they must show the data for this paper to be acceptable for publication.
Author Response
This is an interesting article that makes the point that broadly neutralizing antibodies (BNAbs) against HIV-1 may differ in their ability to neutralize virus and mediate ADCC based on Fc and hinge region characteristics in addition to epitope recognition and binding by the Fab region. Furthermore, the impact of specific Fc and hinge region on antibody functionality may vary depending on the Fab specificity. This is not a surprising finding as many other studies have shown instances of such differences, however this study provides greater detail in a broader survey of BNAb specificities presented with a variety of FC and hinge regions. More importantly, this paper is the first presentation of a novel ADCC assay that may be easier to perform than standard such assays. Unfortunately, there is one major flaw and several minor flaws in the paper that make it unsuitable for publication in its current form.
Thank you for your appreciation of our work.
The major flaw has to do with the description of the new method and is actually noted by the authors themselves in the discussion where they state “This study has limitations. The test used here to measure ADCC do (sic) not reflect the physiological ADCC that occurred in the body, as there are no effector cells like PBMCs or NKs. … further studies are needed to compare our assay with a more physiologic assay using PBMCs or NK cells.”
We fully agree with this comment also highlighted by reviewer 1. We have decided to rephrase the name of the test as ADCC-like assay to better fit the reality. Moreover, as indicated to reviewer 1, we no longer have access to the antibodies used in the study because the company InvivoGen has stopped producing them for us. Our stocks are unfortunately contaminated.
They state in Materials and Methods (2.6 the description of the new assay) that “Ability of HEK-CD16+ or HEK-CD89+ to mimic ADCC were compared to classical NK-based ADCC assay” but then no data or reference to published data for the comparison is given. They further state in Results (3.1) that “We previously defined a new method able to monitor the potential ADCC activity of different mAbs using HEK 293-gp41MSD and CHOgp140 cells as target cells” and give two references for that statement; but neither reference is to a paper that even mentions the new method (one is a reference to the HEK 293-gp41MSD cell line and the other is a totally unrelated article that just happens to be co-authored by three of the authors on this paper). As the first report of a new assay it is unacceptable to not include any data comparing the new assay with standard assays, especially if the authors themselves realize this limits the paper and actually have the data as they appear to indicate.
Indeed, the company InvivoGen indicated to us that they have compared their lines to a classical ADCC activity. However, they were not able to provide us with data despite our request. We therefore decided to modify the manuscript and to name our test ADCC like and to delete the part indicating the comparison with classical ADCC tests.
Another problem with the paper is the data itself, much of which represents trends that do not actually meet statistical significance.
This is indeed true and we have clearly pointed out the lack of significance when this is the case.
Lastly, some of the language in the is awkward and sounds like a non-native speaker of English used google to translate what was written in another language; this paper would benefit from being proofread by someone who is both a native speaker of English and knowledgeable about the science.
We had the text corrected by a native english speaker.
We are very sorry that we cannot provide InvivoGen's data on the comparison with a chromium ADCC test and that we cannot compare it with the HIV antibodies in the study. We hope that the rewording to ADCC-like assay will limit the confusion and narrow our conclusions. However, this ADCC-like assay seems to us to be particularly interesting in the study of the functional properties of antibodies and in particular as a screening assay. This is the purpose of this publication.
Reviewer 3 Report
The study produced IgG1 and IgAs for 2 gp41-directed (or MPER-directed) and 6 gp120-directed HIV bNAbs and then assessed the proteins for neutralization and FcR cross-linking signaling. The authors found differences in bNAb functions, either neutralization or FcR cross-linking, among IgG1 and IgA isotypes. Overall, the data is informative and worth publishing. Below are a few weaknesses to address.
1. A major problem is the use of the general term “ADCC”. As noted, there’s no cell killing measured in this study. Instead, HEK-CD16+ and HEK-CD89+ cells were used, basically measuring FcR cross-linking signaling in these cells. The HEK-CD16+ may be a mimicry for ADCC by CD16+ NK cells, but NK cells do not express CD89. So, it’s confusing to see statements such as IgAs are better than IgGs to mediate ADCC in some cases. Since phagocytes such as monocytes and neutrophils express both CD16 and CD89, HEK-CD16+ and HEK-CD89+ cells could be a mimicry for ADCP, but not necessarily for ADCC. It may be better to report what was measured: FcR cross-linking signaling in HEK-CD16+ and HEK-CD89+ cells.
2. Title: “HIV1” should be “HIV-1”.
3. Methods, “TZMbl” should be “TZM-bl”.
4. “NIH4546” should be “NIH45-46”.
5. Methods, 2.4, 11307 is panel B, not panel C. It would be more accurate to state clade B or clade C, not panels.
6. Figure 2 and Results, 3.2.2, what’s CHOSEC? Is it the same as CHOgp140?
7. Results, 3.3, the section title is wrong.
Author Response
The study produced IgG1 and IgAs for 2 gp41-directed (or MPER-directed) and 6 gp120-directed HIV bNAbs and then assessed the proteins for neutralization and FcR cross-linking signaling. The authors found differences in bNAb functions, either neutralization or FcR cross-linking, among IgG1 and IgA isotypes. Overall, the data is informative and worth publishing. Below are a few weaknesses to address.
Thank you for your appreciation of our study.
- A major problem is the use of the general term “ADCC”. As noted, there’s no cell killing measured in this study. Instead, HEK-CD16+ and HEK-CD89+ cells were used, basically measuring FcR cross-linking signaling in these cells. The HEK-CD16+ may be a mimicry for ADCC by CD16+ NK cells, but NK cells do not express CD89. So, it’s confusing to see statements such as IgAs are better than IgGs to mediate ADCC in some cases. Since phagocytes such as monocytes and neutrophils express both CD16 and CD89, HEK-CD16+ and HEK-CD89+ cells could be a mimicry for ADCP, but not necessarily for ADCC. It may be better to report what was measured: FcR cross-linking signaling in HEK-CD16+ and HEK-CD89+ cells.
We fully agree with this point and as indicated for reviewers 1 and 2 we decided to change the name of our assay to ADCC-like or ADCC-mimmicry assay. ADCC could be mediated by neutrophils with IgA as previously described in the litterature.
- Title: “HIV1” should be “HIV-1”
Ok modified
- Methods, “TZMbl” should be “TZM-bl”
Ok modified
- “NIH4546” should be “NIH45-46”.
Ok modified
- Methods, 2.4, 11307 is panel B, not panel C. It would be more accurate to state clade B or clade C, not panels.
Ok modified as suggested buy the reviewer
- Figure 2 and Results, 3.2.2, what’s CHOSEC? Is it the same as CHOgp140?
Yes exactly this is the two names of the cell line. We modified the text accordingly.
- Results, 3.3, the section title is wrong.
The objectives of the Results 3.3 is to correlate ADCC-like activity and neutralization.
We thank reviewer 3 for these comments and improvement of the manuscript.
Round 2
Reviewer 1 Report
The Authors addressed the reviewer's comments appropriately.